# Surgical procedures for children in the public healthcare sector: a nationwide, facility-based study in Uganda

Mary Margaret Ajiko [1,2] Julia Kressner,[2] Alphonsus Matovu,[2,3] P Nordin,[4,5] Andreas Wladis,[6] Jenny Löfgren [2,7]

¹Surgery, Soroti Regional Referral Hospital, Kampala, Uganda
²Molecular Medicine and Surgery, Karolinska Institute, Stockholm, Sweden
³Surgery, Mubende Regional Referral Hospital, Kampala, Uganda
⁴Department of Surgical and Perioperative Sciences, Umeå University, Umeå, Sweden
⁵Department of Surgery, Östersunds sjukhus, Ostersund, Sweden
⁶Department of Clinical and Experimental Medicine, Linköping University, Linköping, Sweden
⁷Reconstructive Plastic Surgery, Karolinska University Hospital, Stockholm, Sweden

**Correspondence to**
Dr Jenny Löfgren;
jenny.lofgren@ki.se

## ABSTRACT

**Objective** This study investigated the surgical services for children at the highest levels of the public healthcare sector in Uganda. The aim was to determine volumes and types of procedure performed and the patients and the human resource involved.

**Design** The study was a facility-based, record review.

**Setting** The study was carried out at the National Referral Hospital, all 14 regional referral hospitals and 14 general hospitals in Uganda, representing the highest levels of hospital in the public healthcare sector.

**Participants** The subjects were children <18 years who underwent major surgery in the study hospitals during 2013 and 2014.

**Results** The study hospitals contribute with an average annual rate of paediatric surgery at 22.0 per 100 000 paediatric population. This is a fraction of the estimated need. Most of the procedures were performed for congenital anomalies (n=3111, 39.4%), inflammation and infection (n=2264, 28.7%) and trauma (n=1210, 15.3%). Specialist surgeons performed 60.3% (n=4758) of the procedures, and anaesthesia was administered by specialist physician anaesthetists in 11.6% (n=917) of the cases.

**Conclusions** A variety of paediatric surgical procedures are performed in a relatively decentralised system throughout Uganda. Task shifting and task sharing of surgery and anaesthesia are widespread: a large proportion of surgical procedures was carried out by non-specialist physicians, with anaesthesia mostly delivered by non-physician anaesthetists. Reinforcing the capacity and promoting the expansion of the health facilities studied, in particular the general hospitals and regional referral hospitals, could help reduce the immense unmet need for surgical services for children in Uganda.

### Strengths and limitations of this study

► This study included information about surgical procedures carried out in children in 29 hospitals (primary, secondary and tertiary institutions) within the public healthcare sector all over Uganda and therefore adds important information compared with previous research carried out in fewer hospitals or in a smaller geographical area.

► The names of the surgical and anaesthesia providers were captured in the data collection, and this made it possible to find out to what extent task sharing in surgery and anaesthesia for paediatric patients is practised in Uganda.

► The main limitation of the study is the retrospective study design and that the level of detail was limited to that of the logbooks reviewed.

► The data collected is now (2021) 7–8 years old, preceding recent global surgery efforts.

► The study focuses on volumes of surgery and lacks outcomes data, which is a limitation as quantity and quality of surgical services should ideally not be separated.

Children requiring surgical intervention make up a significant proportion of child admissions. Information available suggests that 6%–12% of all paediatric admissions in sub-Saharan Africa are surgical, although the proportion may be higher in some urban areas.[5] In a regional referral hospital (RRHs) in Uganda, a third of the patients admitted to a surgical department during a 3-year period were below 18 years of age.[6] This however is not enough to meet the need for surgical services for children. In a cross-sectional survey performed in Uganda, Sierra Leone, Rwanda and Nepal, 19% of children had a surgical condition, and 62% of these children had at least one unmet surgical need.[3] A study in Uganda found that 7.4% of children were currently suffering from an untreated surgical condition.[7]

Uganda is a sub-Saharan, low-income country with a population of over 20 million

## INTRODUCTION

In recent years, global surgery has become a field of its own with the goal of providing access to safe and affordable surgery for all. In 2017, 1.7 billion children and adolescents lacked access to surgical care.[1] Many childhood surgical conditions can be treated successfully with surgery, while failure to do so may lead to lifelong complications, disability or even death.[2 3] The provision of surgical services for children is highly cost-effective.[4]

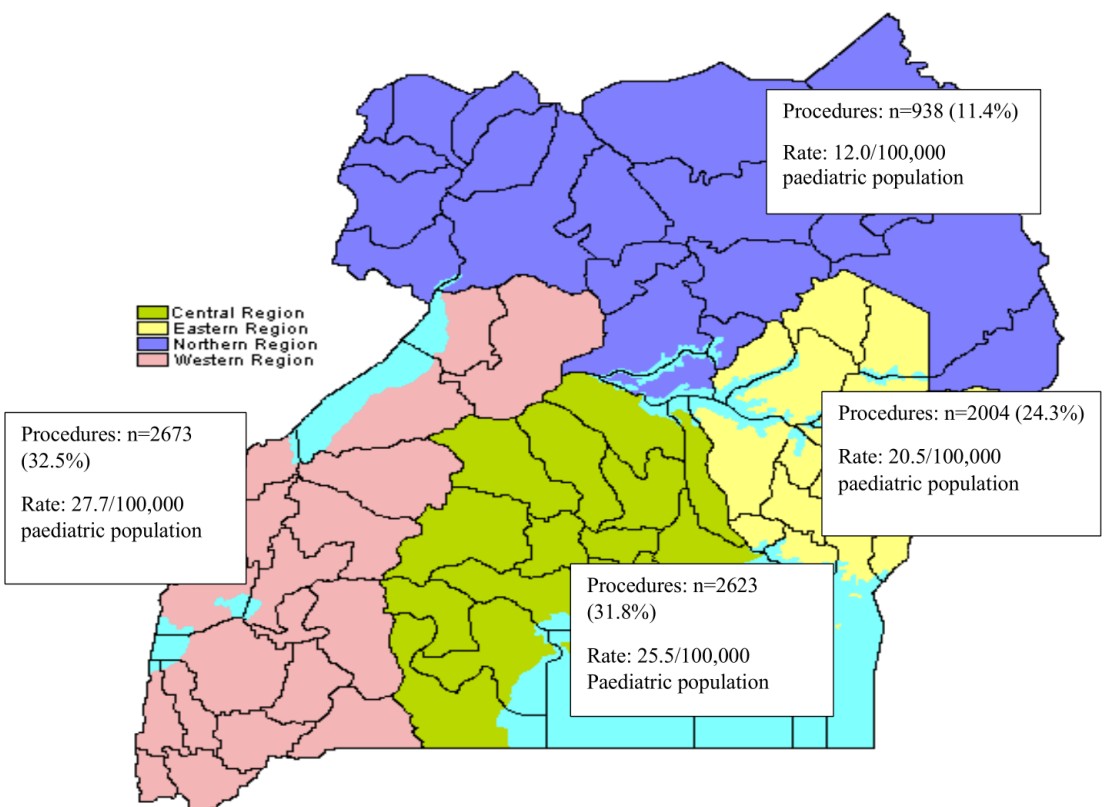

Procedures: n=938 (11.4%)

Rate: 12.0/100,000 paediatric population

Procedures: n=2004 (24.3%)

Rate: 20.5/100,000 paediatric population

Procedures: n=2673 (32.5%)

Rate: 27.7/100,000 paediatric population

Procedures: n=2623 (31.8%)

Rate: 25.5/100,000 Paediatric population

Central Region
Eastern Region
Northern Region
Western Region

**Figure 1** Volumes and rates of paediatric surgery per region in Uganda. **Northern region:** 7 188 139 inhabitants (Apac General Hospital (GH), Nebbi GH, Kitgum GH, Matany GH, Lira RRH, Gulu Regional Referral Hospital (RRH) and Moroto RRH). **Eastern region:** 9 042 422 inhabitants (Lwala GH, Tororo GH, Iganga GH, Soroti RRH, Mbale RRH and Jinja RRH). **Western region:** 8 874 862 inhabitants (Kilembe GH, Kisoro GH, Masindi GH, Itojo GH, Hoima RRH, Fort Portal RRH and Mbarara RRH/University Teaching Hospital). **Central region:** 9 529 227 inhabitants (Entebbe GH, Kalizizo GH, Mityana GH, Masaka RRH, Naguru RRH, Mubende RRH and Mulago NRH/Makerere University Teaching Hospital). Uganda Bureau of Statistics, National Population and Housing Census 2014, https://www.ubos.org/wpcontent/uploads/publications/03_20182014_National_Census_Main_Report. pdf. Total population: 34.6 million with ages 0–17 years old=55.1%.

children.[8] With a birth rate of 5.0 children per women, the paediatric population is rapidly increasing.[9] Although 80% of the Ugandan population lives in rural areas, 95% of surgeons practise in an urban environment.[10] The extent to which the healthcare system can provide surgical services for children at the national level is not known. The objective of the study was to investigate volumes and types of paediatric surgery performed at the highest-level hospitals in Uganda, to relate that to the need for surgery in the paediatric population and to describe the human resource delivering the services.

## METHOD
### Study design
This was a retrospective review of relevant hospital records.

### Study location, study population and study size
Some 29 hospitals geographically distributed throughout Uganda were included. These hospitals represent the highest level of public healthcare and include the National Referral Hospital (NRH) (Mulago), all 14 RRHs and an additional 14 general hospitals (GHs) (figure 1).

Data from two RRHs were incomplete, and they were therefore excluded from the analysis.

The public healthcare system in Uganda is divided into seven levels: from village health teams (level I) to health centres (levels II–IV) and hospitals (levels V–VII). The hospitals are divided into GHs (district) (level V), RRHs (level VI) and the NRH (level VII). All private hospitals are classified as level V hospitals. Uganda has 113 GHs, 38 of which are public-service hospitals. The rest are private and private non-profit hospitals.[11] Each RRH has at least one GH in its catchment area. Due to scarcity of suitable public GHs in the Northern and Eastern regions, two non-governmental organisation GHs were included in the study.

The catchment populations between the GHs, RRHs and NRH overlap as they are meant to provide services of increasing complexity. The GHs and the RRHs were designed for a catchment population of 500 000 people and two million people, respectively, but these hospitals cover much larger populations now due to rapid population growth. GHs have specialists in internal medicine, gynaecology, obstetrics and paediatrics, but not in surgery. Non-specialist medical doctors perform surgical

procedures. Each RRH has a mandate to receive referred patients from the regions they serve, but self-referral from other regions is also permitted. The RRHs employ general surgeons. The NRH covers the entire population for highly specialised services. It is also a teaching facility of Makerere College of Health Sciences. The three paediatric surgeons and the paediatric anaesthetist in Uganda are employed at this institution.

Since 2001, all public healthcare facilities have been officially free of charge,[12] which is why the public healthcare sector was selected for this survey. To describe the surgical activity for children here, the NRH and all the 14 RRHs were included. In addition, one GH per RRH was selected. It was hypothesised that the surgical activity for children in the GHs would be low, as no surgeons are employed in these hospitals. This selection of hospitals allowed comparisons between the three highest levels in the public healthcare sector.

Logbooks from operating theatres for the years 2013 and 2014 at the hospitals studied were photographed, and the information extracted was entered into excel spreadsheets. Data on all major surgical procedures performed on children were included. The logbooks provided, for each hospital, information about children that had undergone a surgical procedure. They included date of procedure, age of the patient, diagnosis, type of surgical procedure performed, names of surgeon and anaesthetist and type of anaesthesia used. Each surgeon's and anaesthetist's training level was thereafter clarified with each hospital. In this study, a child is defined as 'a human being below the age of 18 years', according to the United Nations (UN) Convention on Children's Rights. This is also in keeping with the Ugandan Constitution.[13] In 2014, children aged 0–17 years old represented 55.1% of the population, or 19.1 million children.[14]

A major surgical procedure was defined as 'any procedure conducted in the operating theatre involving the incision, excision, manipulation or suturing of tissue, which usually requires regional or general anaesthesia or profound sedation to control pain'.[15] An anaesthetic assistant/officer was a healthcare provider who had undertaken a 2–3-year course in anaesthesia at a recognised institution and had been certified by the Allied Health Professionals Council of Uganda.[16] A senior house officer was a resident in surgery or anaesthesia. An intern had completed medical school and formally practises under the supervision of a senior colleague. A specialist physician is a doctor who has completed advanced education and training in a specific field of medicine.

## Data analysis
Data were analysed by region and by hospital type using standard descriptive methods. Data are presented to describe the basic characteristics of the patients, the procedures and the human resource providing the anaesthesia and surgical services. Analyses used for group comparisons were Pearson's $t$ test for continuous data and the $\chi^2$ test for proportions and identification of associations between demographical characteristics and indications for surgery. We also categorised gender and age distribution with the number of procedures performed on children for each hospital level. Missing data on the number of procedures performed were handled by multiple imputation. The surgical volumes at the NRH were distributed to the entire population with the relative proportion of 74% to the Central region, 12% to the Western region, 13% to the Eastern region and 1% to the Northern region. This distribution was based on previous research on geographical distribution of paediatric surgical patients presenting to the NRH.[17] The 2014 national population census for Uganda was used to calculate the rate of surgical procedures for children per 100 000 paediatric population for 2014. The paediatric population of Uganda for 2013 was calculated based on the 2014 national census and a population growth of 3.3% per year.[18]

In order to relate the hospital surgical productivity to the need for surgery, a population-based study on paediatric surgical need carried out in Uganda in 2014 was used.[7]

For identification of common diagnoses and procedures, only complete original data were used. Two RRHs were excluded from the analyses; one had no data for the entire study period and the other omitted patient age in the majority of cases, making it impossible to extract all paediatric cases.

## Ethical considerations
Ethical approval was granted by the Makerere University School of Public Health (HDREC 076) and the Uganda National Council for Science and Technology (HS1892). The medical superintendent of each hospital approved collection of data from the theatre logbooks. The retrospective nature of the study meant that the treatment delivered to the patients was not affected. To protect their integrity, patients' names were not included.

## Role of the funding source
The funder of the study had no role in the study design, data collection, data analysis, data interpretation, writing of the article or the decision to submit for publication. All authors had full access to all the data in the study and were responsible for the decision to submit for publication.

## Patient and public involvement
There was no involvement of patients or the public.

## RESULTS
A total of 7886 children who had undergone a major surgical procedure were identified in the theatre records of the 27 hospitals during the study period. After imputation for missing data, this number was 8238 of which 4680 were operated in 2013 and 3558 in 2014. This represented an average annual rate of 22.0 major surgical procedures per 100 000 paediatric population of Uganda in 2013–2014. The rate varied between the four regions in

**Table 1** Surgical need and contribution towards the met need by the study hospitals

| | Need for surgical consultation | Need for surgical intervention | Children operated in study hospitals | Interpretation |
|---|---|---|---|---|
| Proportion of children | 7.4%* | 7.6%–11.8% of children with need for surgical consultation* | 4.9%–7.6% of estimated need for a major surgical procedure | The 8238 children operated correspond to 4.9%–7.6% of the estimated need for a major surgical procedure in the paediatric population of Uganda in 2014 |
| Number of children | 1 413 400* (7.4% of 19.1 million paediatric population in 2014) | 107 418–166 781* | 8238 | Another 99 000–159 000 surgical procedures are needed to meet the need for major surgical interventions in children in Uganda |
| Rate, n per 100 000 paediatric population | 7400* | 562–873* | 22.0 | The surgical rate contributed by the study hospitals is a fraction of the estimated need |

*Butler *et al*.[7]

Uganda: from 12.0 per 100 000 children in the Northern region to 27.7 per 100 000 children in the Central region (figure 1). There was considerable variation in surgical volumes within the RRH and GH groups, and the most productive RRH performed 21 times more surgical procedures on children than the least productive.

A population-based study in Uganda previously investigated the prevalence of surgical conditions and the need for surgical consultation and surgical intervention among children in Uganda.[7] In relation to the findings of this previous study, the study hospitals in the present study contributed with surgical volumes corresponding to 4.9%–7.6% of the need for major surgical procedures in children (table 1).

The basic characteristics of the patients and the surgical and anaesthesia providers are shown in table 2. Of the children, 43.4% (n=3422) were below 5 years of age, and 60.8% (n=4797) were boys. In online supplemental appendix 1, the age distribution by hospital level is visualised. Most of the procedures (n=4758, 60.3%) were performed by specialists in general surgery or specialists in other surgical specialities (paediatric surgery; gynaecology; orthopaedic surgery; cardiothoracic surgery; plastic surgery; neurosurgery; ear, nose and throat (ENT) surgery; and urology). Overall, surgical residents performed 14.5% (n=1145) of the procedures, and other non-specialist physicians performed 20.5% (n=1616). Interns carried out 3.0% (n=237). Anaesthesia was rarely administered by a physician anaesthetist (n=917, 11.6%) or physician anaesthetic resident (n=818, 10.4%). The majority of the patients were anaesthetised by an anaesthetic assistant or anaesthetic officer (n=5469, 69.4%).

The surgical procedures were performed by 275 medical doctors with different levels of qualification: 103 specialist surgeons in different disciplines of whom three were paediatric surgeons, 57 resident surgeons in different surgical disciplines and 115 medical officers without specific training in surgery. Of the specialists, 40 performed surgery at the NRH and 61 at the RRH. Only two specialist surgeons performed surgical procedures at a GH. Anaesthesia was administered by 18 physician anaesthetists and 99 anaesthetic officers. Of the physician anaesthetists, three worked at an RRH and 15 at the NRH. No physician anaesthetist worked at a GH.

Indications for the surgical procedures are presented in table 3. The most common was related to congenital anomalies (n=3111, 39.4%), infectious and inflammatory conditions (n=2264, 28.7%) and trauma (n=1210, 15.3%). Table 4 shows the surgical procedures that were carried out. Abdominal and colorectal surgical procedures were most common, followed by orthopaedic and ENT procedures. The types of procedure varied between the different levels of hospitals, and in the GH, hernia surgery was the most common. Surgery for congenital heart conditions (n=128, 1.6%) was only performed at the NRH where there is a heart institute run by cardiothoracic surgeons. Neurosurgeons performed all neurosurgical procedures including ventriculoperitoneal shunts, spina bifida repair, brain abscess drainage, craniotomy and meningocele repair. Such procedures were carried out at the NRH (Mulago Hospital) and some RRHs.

## DISCUSSION

The study hospitals in the present study contributed with an average annual surgical rate of 22.0 major surgical procedures per 100 000 paediatric population in Uganda in 2013 and 2014. The patients were often below 5 years old, and the surgical procedures were mainly carried out for congenital anomalies, inflammatory and infectious conditions and trauma. Task sharing and task shifting were widely practised as non-specialists frequently performed surgical procedures and as non-physician clinicians administered anaesthesia in most of the cases.

A considerable variation in the number and rate of major procedures performed at the different hospitals, and within the four regions of the country, was seen. The Northern region was the most underserved,

**Table 2** Basic characteristics of patients and human resource performing surgery for children per hospital type

| | General hospital (GH), n=1276 | Regional referral hospital (RRH), n=4475 | National Referral Hospital (NRH), n=2135 | Total n=7886 | P value |
|---|---|---|---|---|---|
| **Sex, n (%)** | | | | | |
| Boy | 945 (74.1) | 2516 (56.2) | 1336 (62.6) | 4797 (60.8) | <0.001* |
| Girl | 324 (25.4) | 1620 (36.2) | 756 (35.4) | 2700 (34.2) | |
| Not indicated | 7 (0.5) | 339 (7.6) | 43 (2) | 389 (5.0) | |
| **Age distribution (years)** | | | | | |
| Mean (SD) | 7.4 (5.5) | 7.1 (5.5) | 5.8 (5.7) | 5.3 (4.7) | <0.001* |
| <5 | 488 (38.2) | 1790 (40.0) | 1144 (53.6) | 3422 (43.4) | |
| >5 | 777 (60.9) | 2677 (59.8) | 980 (45.9) | 4434 (56.6) | |
| Not indicated | 11 (0.9) | 8 (0.2) | 11 (0.5) | 30 (0.4) | |
| **Level of training of surgical provider** | | | | | |
| Specialist surgeon (including three paediatric surgeons) | 324 (25.4) | 3063 (68.4) | 1371 (64.2) | 4758 (60.3) | <0.001* |
| Resident | 0 | 437 (9.8) | 708 (33.2) | 1145 (14.5) | |
| Medical officer | 904 (70.8) | 700 (15.6) | 12 (0.6) | 1616 (20.5) | |
| Intern | 0 | 232 (5.2) | 5 (0.2) | 237 (3.0) | |
| Not indicated | 48 (3.8) | 43 (1.0) | 39 (1.8) | 130 (1.6) | |
| **Level of training of anaesthesia provider** | | | | | |
| Physician anaesthetists (one paediatric anaesthesiologist) | 0 | 807 (18.0) | 110 (12.0) | 917 (11.6) | <0.001* |
| Residents in anaesthesia | 0 | 163 (3.6) | 655 (30.7) | 818 (10.4) | |
| Self/by surgeon local anaesthesia | 47 (3.7) | 162 (3.6) | 10 (0.5) | 219 (2.8) | |
| Anaesthetic assistant or officer | 1190 (93.3) | 3132 (70) | 1147 (53.7) | 5469 (69.4) | |
| Not indicated | 39 (3.1) | 211 (4.7) | 213 (10) | 463 (5.9) | |
| **Type of anaesthesia** | | | | | |
| General anaesthesia | 1097 (86.0) | 3811 (85.2) | 1913 (89.6) | 6821 (86.5) | <0.001* |
| Spinal anaesthesia | 66 (5.2) | 186 (4.2) | 82 (3.8) | 334 (4.2) | |
| Local anaesthesia±sedation | 106 (8.3) | 161 (3.6) | 64 (3.0) | 331 (4.2) | |
| Not indicated | 7 (0.5) | 317 (7.1) | 76 (3.6)) | 400 (5.1) | |

*Represents p values of comparison of gender, age, level of training of surgical and anaesthesia provider and type of anaesthesia between GH, RRH and the NRH, respectively.

where hospitals are few and the distance to the NRH is greatest. This is similar to the results of a previous study in Uganda, which found a positive correlation between burden of paediatric surgical disease and distance to the nearest healthcare unit offering surgery for children.[19] The majority of the Ugandan population lives in a rural setting with very limited financial resources.[6 20] Transport costs are likely to pose a barrier to surgical care for children there. This is confirmed by previous research in low- and middle-income countries showing that cost and travelling distance are barriers to surgical care access.[21] Decentralised services are therefore needed.

The Global Initiative for Children's Surgery developed the Optimal Resources for Children's Surgical Care, published in 2019. This document stratifies by level of hospital what services and human resource should be available at each level. The recommendations are in line with what is demonstrated in the present study. On GH level, intermediate surgical services, including hernia repair and surgical management of acute abdominal and traumatic conditions, should be provided.[22] In the GHs in the present study, hernia surgery represented the vast majority of procedures performed, followed by other abdominal and orthopaedic procedures. These procedures were carried out on all three levels, but in addition, the RRHs and NRH performed procedures of increasing complexity including surgical subspecialties.

For essential surgical care to be accessible for all children in Uganda, service delivery in both GHs and RRHs should be further promoted. It is encouraging to see that

**Table 3** Indications for surgical procedures for children in the public healthcare sector in Uganda (2013–2014)

| | General hospital 1276 patients, n (%) | Regional referral hospital 4475 patients, n (%) | National Referral Hospital 2135 patients, n (%) | Total 7886 patients, n (%) |
|---|---|---|---|---|
| **Abdominal and colorectal symptoms and conditions (n=3774)** | | | | |
| Groin hernia and hydrocoele | 499 (39.1) | 650 (14.5) | 120 (5.6) | 1269 (16.1) |
| Peritonitis | 49 (3.8) | 318 (7.1) | 106 (5.0) | 473 (6.0) |
| Umbilical hernia | 113 (8.9) | 185 (4.1) | 72 (3.4) | 370 (4.7) |
| Gut perforation | 31 (2.4) | 240 (5.4) | 61 (2.9) | 332 (4.2) |
| Imperforate anus | 13 (1.0) | 109 (2.4) | 163 (7.6) | 285 (3.6) |
| Intussusception | 18 (1.4) | 130 (2.9) | 126 (5.9) | 274 (3.5) |
| Intestinal obstruction | 47 (3.7) | 124 (2.8) | 89 (4.2) | 260 (3.2) |
| Appendicitis | 24 (1.9) | 60 (1.3) | 42 (2.0) | 126 (1.6) |
| Abdominal injuries (blunt and penetrating) | 10 (0.8) | 62 (1.4) | 37 (1.7) | 109 (1.4) |
| Ruptured spleen | 12 (0.9) | 60 (1.3) | 18 (0.8) | 90 (1.1) |
| Colostomy/closure of colostomy | 10 (0.8) | 44 (1.0) | 20 (0.9) | 74 (1.0) |
| Others | 19 (1.5) | 74 (1.7) | 19 (0.9) | 112 (1.4) |
| **Orthopaedic conditions (n=1264)** | | | | |
| Orthopaedic trauma | 62 (4.9) | 366 (8.2) | 224 (10.5) | 652 (8.2) |
| Chronic osteomyelitis and septic arthritis | 137 (10.7) | 254 (5.7) | 82 (3.8) | 473 (6.0) |
| Clubfoot | 61 (4.8) | 53 (1.2) | 25 (1.2) | 139 (1.8) |
| *Total* | 260 (20.4) | 673 (15.0) | 331 (15.5) | 1264 (16.0) |
| **Ear, nose and throat conditions (n=721)** | | | | |
| Tonsillitis/enlarged adenoids/foreign body, goitre | 1 (0) | 548 (12.2) | 172 (8.0) | 721 (9.1) |
| **Urology (n=444)** | | | | |
| Cryptorchidism | 30 (2.4) | 63 (1.4) | 12 (0.6) | 105 (1.3) |
| Testicular torsion | 8 (0.6) | 41 (0.9) | 36 (1.7) | 85 (1.1) |
| Others | 7 (0.5) | 138 (3.1) | 107 (5.0) | 251 (3.2) |
| **Neurosurgical conditions (n=316)** | | | | |
| Spina bifida, hydrocephalus, meningocele, subdural haemorrhage, trauma | 0 (0.0) | 135 (3.0) | 181 (8.5) | 316 (4.0) |
| Congenital heart disease (n=122) | 0 (0.0) | 0 (0.0) | 122 (5.7) | 122 (1.5) |
| **Plastic surgical conditions (n=503)** | | | | |
| Cleft lip and palate | 27 (2.1) | 107 (2.4) | 10 (0.5) | 144 (1.8) |
| Burns (acute and secondary surgery) | 50 (3.9) | 243 (5.4) | 66 (3.1) | 359 (4.6) |
| Malignancies (n=92) | 1 (0.1) | 68 (1.5) | 24 (1.1) | 92 (1.1) |
| Others*, (n=896) | 67 (5.3) | 630 (14.1) | 212 (9.9) | 896 (11.4) |
| Not indicated (n=51) | 7 (0.5) | 36 (0.8) | 8 (0.4) | 51 (0.6) |

*Indications with ≤1% were grouped together under 'others'. Some patients had more than one indication for surgery; hence, total number of conditions and symptoms exceeds number of patients.

surgery for children was delivered in these hospitals even if at rates that represent only a fraction of the need. The more productive hospitals in each group could serve as models for the less productive ones. A first goal would be for all hospitals on the same level to deliver the same services and similar volumes of surgery. The present study did not investigate the outcomes after surgery, and

this should be evaluated in future research. Volume and quality represent two sides of the same coin, and capacity building must take both into consideration.

Significantly, more boys than girls were recorded in this study. Several of the most common diagnoses requiring surgery were gender-specific, or over-represented, in men. For example, inguinal hernia, which was the most

**Table 4** Paediatric surgical procedures by hospital level in Uganda (2013–2014)

| Procedure | General hospital, 1276 patients, n (%) | Regional referral hospital, 4475 patients, n (%) | National Referral Hospital, 2135 patients, n (%) | Total n=7886 patients, n (%) |
|---|---|---|---|---|
| Abdominal and colorectal surgery (n=4559 procedures) | | | | |
| Laparotomy | 179 (14.0) | 990 (22.1) | 453 (21.2) | 1622 (20.6) |
| Herniotomy/herniorrhaphy/ hydrocelectomy | 591 (46.3) | 652 (14.6) | 121 (5.7) | 1364 (17.3) |
| Colostomy (creating/closure) | 17 (1.3) | 206 (4.6) | 175 (8.2) | 398 (5.0) |
| Mayo's repair of umbilical hernia | 113 (8.9) | 179 (4.0) | 68 (3.2) | 360 (4.6) |
| Resection and anastomosis | 12 (0.9) | 149 (3.3) | 83 (3.9) | 244 (3.1) |
| Appendicectomy | 30 (2.4) | 58 (0.8) | 45 (2.1) | 133 (1.7) |
| Splenectomy | 7 (0.5) | 73 (1.6) | 13 (0.6) | 93 (1.2) |
| Other abdominal procedures (Swenson pull through, posterior sagittal anorectoplasty, Ramtsaedt's procedure) | 0 (0.0) | 195 (4.4) | 150 (7.0) | 345 (4.4) |
| Orthopaedic procedures (n=1212 procedures) | | | | |
| Fracture surgery (internal and external fixations), amputations | 72 (5.6) | 452 (10.1) | 154 (7.2) | 678 (8.6) |
| Sequestrectomy | 115 (9.0) | 141 (3.2) | 54 (2.5) | 310 (3.9) |
| Corrective procedures for clubfoot | 61 (4.8) | 63 (1.4) | 25 (1.2) | 149 (1.9) |
| Arthrotomy | 9 (0.7) | 43 (1.0) | 23 (1.1) | 75 (1.0) |
| Ear, nose and throat (n=594 procedures) | | | | |
| Adenotonsillectomy | 0 (0) | 292 (6.5) | 115 (5.4) | 407 (5.2) |
| FB removal, bronchoscopy, others | 2 (0.2) | 134 (3.0) | 51 (2.4) | 187 (2.4) |
| Urology (n=429) | | | | |
| Orchidopexy and orchidectomy | 41 (3.2) | 115 (2.6) | 34 (1.6) | 190 (2.4) |
| Reconstructive surgery (hypospadia, epispadia, urethral valve) | 4 (0.3) | 130 (3.0) | 105 (5.0) | 239 (3.0) |
| Cardiothoracic and vascular surgery (n=128 procedures) | 0 (0) | 0 (0) | 128 (6.0) | 128 (1.6) |
| Plastic surgery (n=293 procedures) | | | | |
| Cleft lip and palate | 31 (2.4) | 132 (2.9) | 50 (2.3) | 213 (2.7) |
| Skin grafting | 22 (1.7) | 43 (1.0) | 15 (0.7) | 80 (1.0) |
| Neurosurgery (n=285 procedures) | | | | |
| Neurosurgery procedures (VP shunt, drainage of brain abscess, craniotomy, repair of meninges) | 0 (0) | 122 (2.7) | 163 (7.6) | 285 (3.6) |
| Other procedures (n=782 procedures) | 92 (7.2) | 575 (12.8) | 115 (5.4) | 782 (9.9) |
| Not indicated (n=16 procedures) | 0 (0) | 16 (0.4) | 0 (0) | 16 (0.2) |

*Procedures with ≤1% were grouped under 'others'. Some patients had more than one surgical procedure performed; hence, total number of procedures exceeds number of patients.

FB, foreign body; VP, ventriculoperitoneal .

common indication for surgery in the GHs, is more common in boys than in girls.[23] Female-specific diagnoses such as gynaecological conditions were rare. This study could not determine whether boys have better access to surgery than girls, but future research into gender as a barrier to surgical care in children could investigate this further.

Task sharing and task shifting of surgical duties and anaesthesia delivery were routinely practised in the study hospitals. Task sharing with medical doctors without specific training in surgery is common in many countries, but the safety and effectiveness as well as the training available for this important group of surgical providers are not well understood.[24] Ways in which the highest possible quality of paediatric surgical services can be achieved and maintained must be the subject of future research. Standardisation of the surgical training and support and monitoring of non-surgeons for the the most common surgical conditions should be prioritised. The anaesthetic officers in Uganda have formal training, and their duties are probably better regulated than those of non-specialist medical doctors. Nonetheless, regular monitoring of the surgical and anaesthetic work forces and the institutions in which they work is an essential component of the provision of high-quality and cost-effective surgical services for children. An additional important role of the healthcare system is to promote prevention: many surgical conditions can be prevented. Antenatal care and vaccinations are carried out in the study hospitals, and health information and child safety issues could be discussed with parents during such visits.

Surgery has an important place in several of the health targets in the UN's Sustainable Development Goal 3.[25] Uganda has a national plan for service standards, and service delivery standards for the health sector include a package for surgical, obstetric and anaesthesia plans in scaling up surgical volumes and access to surgical care. However, a defined paediatric surgical package is not included.[26] To reduce the death toll and disability caused by surgical conditions in the paediatric population, the healthcare system must be improved in order to deliver safe surgery and prevention strategies. Advancing and promoting the service delivery already in place could be part of the solution to reach the goal of surgical services of quality for all children with surgical conditions.

## Strengths and limitations

This study set out to investigate the situation for surgery for children in the public healthcare system in Uganda and to describe patients and their surgical providers at the three highest levels of hospitals. The study has several limitations. Due to limited resources, not all hospitals in the country could be visited, and therefore, overall volumes and surgical rates could not be calculated. The hospitals studied were the healthcare facilities first in line for the implementation of new policies of intervention, where Ministry of Health policy as well as increased funding would have a direct effect. Our choice of hospitals

was thus relevant regarding current capacity and surgical volume, providing baseline information that is essential in the future evaluation of the effects of investment and interventions made to enhance and expand paediatric surgical services.

Given the retrospective collection of data in this study, the level of detail is limited to that of the logbooks reviewed. Information on outcome after paediatric surgery in relation to level of hospital and level of training of the surgical and anaesthesia providers would be very valuable. Where data on volumes of surgery performed during the study period were missing, in the selected hospitals, these were imputed as the best option. However, the data sufficed to provide important information about the state of surgery for children in Uganda.

The present data were collected in 2015–2017, and since then, no significant investments have been made for improving surgical services for children on the national level. There is still (2021) only one paediatric anaesthetist, three registered paediatric surgeons and two hospitals with dedicated paediatric operating theatres. The findings should thus still apply today.

**Acknowledgements** The authors acknowledge the invaluable assistance of the hospital directors and staff, which made the study possible. The authors would like to thank Anna Peterson, William Yu and Andreas Vikholm for their important contribution in data collection and management. We also thank Dr Matteo Bottai for statistical assistance.

**Contributors** Study conception and design: JL, AW, MMA and PN. Data acquisition: MMA, JL, JK and AM. Analysis and data interpretation: MMA, JL, JK, AW, PN and AM. Drafting of the manuscript: MMA. Critical revision: MMA, JL, AW, PN, AM and JK.

**Funding** This work was supported through a grant from the Department of Public Health and Clinical Medicine, Unit of Research, Education and Development, Östersund, Umeå University, and by the Swedish Research Council.

**Map disclaimer** The depiction of boundaries on the map(s) in this article does not imply the expression of any opinion whatsoever on the part of BMJ (or any member of its group) concerning the legal status of any country, territory, jurisdiction or area or of its authorities. The maps are provided without any warranty of any kind, either expressed or implied.

**Competing interests** None declared.

**Patient consent for publication** Not required.

**Provenance and peer review** Not commissioned; externally peer reviewed.

**Data availability statement** Data are available on reasonable request. Data will be available on reasonable request.

**ORCID iDs**
Mary Margaret Ajiko http://orcid.org/0000-0002-8079-3842
Jenny Löfgren http://orcid.org/0000-0001-5884-0369

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
