## [Reviewer comments · BMJ Open]

ARTICLE DETAILS

TITLE (PROVISIONAL)	Surgical procedures for children in the public health care sector – a nationwide, facility-based study in Uganda
AUTHORS	Ajiko, Mary Margaret; kressner, Julia; Matovu, Alphonsus; Nordin, P.; Wladis, Andreas; Löfgren, Jenny

VERSION 1 – REVIEW

REVIEWER	Meyer, Heidi University of Cape Town, Anaesthesia & Perioperative Medicine
REVIEW RETURNED	11-Feb-2021

GENERAL COMMENTS	Paediatric surgery in Uganda – a nationwide study on volumes, geographic distribution and patient characteristics I would like to commend the authors on their effort on the extensive revision of this manuscript. As I have mentioned previously the provision of surgical services to children in Africa is such an important topic. I would like to ask the authors for clarity on a few points: The results section text quotes “Surgery for congenital heart conditions (n=128, 1.6%) was only performed at the NRH where there is a heart institute run by cardiothoracic surgeons.” This does not match the data in table 3.? Table 3. I am not clear as to: Why liver disease, burst abdomen, pyloric stenosis, and big spleen disease (?) are not included within the abdominal and colorectal section? Or goitre under ENT? Or bladder calculi under urology? I think it would be worth separating “acute burns” vs “contractures”. The anaesthetic and surgical implications are quite different between these.
---

REVIEWER	Ameh, Emmanuel National Hospital, Abuja, Nigeria
REVIEW RETURNED	07-Mar-2021

GENERAL COMMENTS	I congratulate the authors for attempting to evaluate surgical services for children at higher level public referral hospitals in Uganda. However, I have the following concerns:
---

	1. The retrospective data is old and redundant, dating back to 6-7 years. 2. Simply studying Logbooks of 2013 and 2014 would not give a clear picture of the organisation and delivery of surgical services for children. 3. Without outcome information, the data presented is of limited value. Overall, it would be more helpful if the authors presented an up to date data and not just 2 years, dating back to 6-7 years ago. Simply counting numbers is of limited value. The manuscript does not show anything new or different from what is already known and published about surgery for children in Uganda.
--	--

VERSION 1 – AUTHOR RESPONSE

Reviewer: 1

Dr. Heidi Meyer, University of Cape Town

Comments to the Author:

Paediatric surgery in Uganda – a nationwide study on volumes, geographic distribution and patient characteristics

I would like to commend the authors on their effort on the extensive revision of this manuscript. As I have mentioned previously the provision of surgical services to children in Africa is such an important topic.

Dear Dr Meyer,

Thank you for this positive note and for reviewing the manuscript once more!

I would like to ask the authors for clarity on a few points:

The results section text quotes “Surgery for congenital heart conditions (n=128, 1.6%) was only performed at the NRH where there is a heart institute run by cardiothoracic surgeons.” This does not match the data in table 3.?

Table 3.

I am not clear as to:

Why liver disease, burst abdomen, pyloric stenosis, and big spleen disease (?) are not included within the abdominal and colorectal section? Or goitre under ENT? Or bladder calculi under urology? The table has been updated accordingly.

I think it would be worth separating “acute burns” vs “contractures”. The anaesthetic and surgical implications are quite different between these.

This is indeed correct, but in the data, it was not always possible to know if the procedures were done for acute burns or as secondary reconstruction. Therefore, these remain grouped together.

Reviewer: 2

Dr. Emmanuel Ameh, National Hospital, Abuja, Nigeria

Comments to the Author:

I congratulate the authors for attempting to evaluate surgical services for children at higher level

public referral hospitals in Uganda. However, I have the following concerns:

Dear Prof Ameh,
Thank you for appreciating the work.

1. The retrospective data is old and redundant, dating back to 6-7 years.

We are aware that the data is aging. The plan is to follow up with a new study to evaluate if there have been changes over time. Our experience working in Uganda is not that there have been any major efforts to improve the situation for paediatric surgery on national level since the time of the study. This remains to be evaluated, however.

2. Simply studying Logbooks of 2013 and 2014 would not give a clear picture of the organisation and delivery of surgical services for children.

Information from log books does not give us the whole picture for paediatric surgery in Uganda. We believe that it gives important insight, however. What should be done is often not the same as what is done in reality. All regional referral hospitals should have employed general surgeons and these should be able to provide basic paediatric surgical procedures. In the present study we see that practices vary. We also demonstrate to what extent task sharing to general surgeons and non-specialist medical doctors is practiced in the public health care system. This has important implications for training and monitoring.

The present study is one of 4 for the PhD student, Sr Ajiko. The other projects investigate paediatric surgery from other angles including epidemiology, health care seeking behaviour and finally a RCT on paediatric groin hernia. The present study provides important background and baseline information for the remaining studies.

3. Without outcome information, the data presented is of limited value.

Overall, it would be more helpful if the authors presented an up to date data and not just 2 years, dating back to 6-7 years ago. Simply counting numbers is of limited value. The manuscript does not show anything new or different from what is already known and published about surgery for children in Uganda.

We agree that outcomes are very important A registry that collects data on paediatric surgery and outcomes is in place in a few selected hospitals, but not all 29 hospitals included in the present study. An automated, electronic system covering all hospitals could be a way forward in the future.

We politely disagree that the study adds nothing new about paediatric surgery in Uganda. Previous studies have included fewer hospitals. Where surgery is performed, and by whom, is important new knowledge that can be used for surgical and anaesthesia training interventions, and that will be important for monitoring and evaluation of services.

VERSION 2 – REVIEW

REVIEWER	Meyer, Heidi University of Cape Town, Anaesthesia & Perioperative Medicine
REVIEW RETURNED	14-Jun-2021

GENERAL COMMENTS	Paediatric surgery in Uganda – a nationwide study on volumes, geographic distribution and patient characteristics Thank you for the opportunity to review this revised manuscript.
---

	I would like to ask the authors for clarity on the following point: Table 3. Indicates that 3 children required surgery for congenital heart disease at the Regional Hospital Level, although cardiac surgery was only performed at the NRH? The authors have addressed the limitations of the data, including the historical nature of the data. I feel that this is still valuable and detailed information on the provision of paediatric surgery in Uganda.
--	--

VERSION 2 – AUTHOR RESPONSE

Reviewer: 1

Dr. Heidi Meyer, University of Cape Town

Comments to the Author:

Paediatric surgery in Uganda – a nationwide study on volumes, geographic distribution and patient characteristics

Thank you for the opportunity to review this revised manuscript.

Dear Dr Meyer,

Thank you for your valuable input towards this manuscript. We have appreciated it a lot and believe that the manuscript has improved thanks your review.

I would like to ask the authors for clarity on the following point:

Table 3. Indicates that 3 children required surgery for congenital heart disease at the Regional Hospital Level, although cardiac surgery was only performed at the NRH?

Thank you for observing this. For some children, more than one condition/diagnosis/indication was noted in the log books. In the case of these three children, a cardiac condition was reported along with the condition that the child was operated for. Most likely as it is important for anaesthesia. The first child, a 5 months old child, also had a pleural effusion and received a chest tube. The second child got a colostomy due to recto-vestibular fistula. The third child had surgery for an obstructed hernia.

We have removed the cardiac indication for these three children since it was not the reason for the surgical procedure that they underwent.

The authors have addressed the limitations of the data, including the historical nature of the data. I feel that this is still valuable and detailed information on the provision of paediatric surgery in Uganda.

VERSION 3 – REVIEW

REVIEWER	Meyer, Heidi University of Cape Town, Anaesthesia & Perioperative Medicine
REVIEW RETURNED	25-Jun-2021
GENERAL COMMENTS	I have no further comments and believe the article is acceptable for publication.